



# Aqueous phase behavior of glyoxal and methylglyoxal observed with carbon and oxygen K-edge X-ray absorption spectroscopy.

Georgia Michailoudi[1], Jack J. Lin[1], Hayato Yuzawa[2], Masanari Nagasaka[2], Marko Huttula[1],
Nobuhiro Kosugi[2], Theo Kurtén[3], Minna Patanen[1], and Nønne L. Prisle[1]

[1]Nano and Molecular Systems Research Unit, University of Oulu, P.O. Box 3000, FI-90014 Oulu, Finland
[2]Institute for Molecular Science, Myodaiji, Okazaki 444-8585, Japan
[3]Department of Chemistry and Institute for Atmospheric and Earth System Research (INAR), University of Helsinki, P.O. Box 55, FI-00014 Helsinki, Finland

**Correspondence:** N. L. Prisle (nonne.prisle@oulu.fi)

**Abstract.** Glyoxal (CHOCHO) and methylglyoxal ($CH_3C(O)CHO$) are well-known components of atmospheric particles and their properties can impact atmospheric chemistry and cloud formation. To get information on their hydration states in aqueous solutions and how they are affected by addition of inorganic salts (sodium chloride (NaCl) and sodium sulfate ($Na_2SO_4$)), we applied carbon and oxygen K-edge X-ray absorption spectroscopy (XAS) in transmission mode. The recorded C K-edge spectra

show that glyoxal is completely hydrated in the dilute aqueous solutions, in line with previous studies. For methylglyoxal, we identified, supported by quantum chemical calculations, not only C-H, C=O and C-OH bonds, but also fingerprints of C-OH($CH_2$) and C=C bonds. This implies the presence of both mono- and dihydrated forms of methylglyoxal, as well as products of aldol condensation, and enol tautomers of the monohydrates. The addition of salts was found to introduce only very minor changes to absorption energies and relative intensities of the observed absorption features, indicating that the organic-inorganic

interactions at the studied concentrations are not strong enough to affect the spectra in this work. The identified structures of glyoxal and methylglyoxal in aqueous environment support the uptake of these compounds to the aerosol phase in the presence of water and their contribution on secondary organic aerosol formation.

## 1 Introduction

Aerosol particles in the atmosphere have several important effects on atmospheric chemistry and climate. They interact with

solar radiation and participate in cloud formation. The lack of firm constraints on aerosol-cloud interactions translates into the largest single source of uncertainty in predictions of future climate (IPCC, 2013). Some of the processes which contribute prominently to this uncertainty are the formation of secondary organic aerosol (SOA) from volatile precursors in the atmosphere and the aqueous phase interactions of organic compounds impacting aerosol hygroscopic growth and cloud formation potential. Previous work has revealed a large potential impact of interactions between organic and inorganic compounds on

their gas–particle partitioning and water uptake properties (Kurtén et al., 2014; Hansen et al., 2015). However, such interactions, including *salting effects*, i.e. non-ideal solute–solute interactions in solution phase, are currently poorly described on





the molecular level, and have been characterized theoretically or experimentally only for a limited number of atmospherically relevant systems.

Glyoxal (CHOCHO) and methylglyoxal (CH$_3$C(O)CHO) are two $\alpha$-dicarbonyl compounds which are produced for ex-
ample in the gas-phase oxidation of isoprene (Stavrakou et al., 2009) and they can be lost in the gas phase by photochemical reactions (Chen et al., 2000) and oxidation by OH radical (Tyndall et al., 1995). These small molecules have high pure-compound saturation vapor pressures, but are nevertheless known to make significant contributions to SOA formation, being comparable to the contribution of more chemically complex compounds (Fu et al., 2008; Stavrakou et al., 2009). A key to the atmospheric importance of $\alpha$-dicarbonyls is their versatile reactivity in the condensed-phase. Previous studies have suggested
reaction partners for irreversible condensed-phase reactions of glyoxal (and to a lesser extent, methylglyoxal) under several experimental conditions (photochemistry, different particle size and amount of water) including organic and inorganic acids (Surratt et al., 2007), amines (De Haan et al., 2009b; Kliegman and Barnes, 1970), ammonia (Nozière et al., 2009) and amino acids (De Haan et al., 2009a). Aqueous phase glyoxal and methylgyoxal can be involved in two-step hydration reactions as shown in Fig. 1 (Malik and Joens, 2000; Kua et al., 2008; Nemet et al., 2004; Krizner et al., 2009; Kroll et al., 2017). Glyoxal
has two aldehyde groups, which can yield diol and tetrol species upon uptake of water. Methylglyoxal has two carbonyl groups, one aldehyde and one ketone. For glyoxal, both hydration steps have large equilibrium constants (Table 1), while for methyl-glyoxal, the hydration constant K$_{hyd1}$ of the first step in the aldehyde group (Fig. 1) is very large but the second step of the hydration in the ketone group has lower value (Nemet et al., 2004; Krizner et al., 2009). Hydrate formation has been reported to increase the effective Henry's law constant of glyoxal by almost five orders of magnitude (Ervens and Volkamer, 2010;
Wasa and Musha, 1970), and methylgyoxal is likely to display similar behavior with a calculated equilibrium constant for the monohydrate formation reaction above 1000. Upon hydration, glyoxal and methylglyoxal can also react with themselves to form dimers and oligomers (Ervens and Volkamer, 2010; Liggo et al., 2005; Kroll et al., 2005; Wang et al., 2010; Galloway et al., 2009). Both the hydrates and dimers/oligomers are covalently bound species, not hydrogen-bonded clusters.



**Figure 1.** Formation of mono- and dihydrated glyoxal (first row) and methylglyoxal (second row). The values of equilibrium constants, K$_{hyd1}$ and K$_{hyd2}$, are shown in Table 1.

In this work, we have used synchrotron radiation (SR) based soft X-ray absorption spectroscopy (XAS) combined with quan-
tum chemical calculations, to study hydrate formation of glyoxal and methylglyoxal in aqueous solutions, and their aqueous





phase interactions with various salts commonly found in atmospheric aerosol. The high brightness and element selectivity, in terms of core level excitation energy and energy resolution makes SR a convenient tool to study molecular level interactions in aqueous phase using absorption spectroscopy.

## 2 Methods

### 2.1 Sample preparation

Aqueous solutions of each of the organic compounds, glyoxal, methylglyoxal and glycerol ($C_3H_8O_3$), were prepared freshly before each experiment. Glycerol solution is used for comparison with the glyoxal and methylglyoxal solutions because it has similar structure to their hydrated forms. The aqueous solutions were prepared using water purified by Sartorius arium® pro UV system (18.2 MΩ). Methylglyoxal was purchased from Sigma-Aldrich and all the other chemicals from Wako. All chemicals were used without further treatment. Glyoxal and methylglyoxal were delivered as 40 wt% aqueous solutions, while glycerol was a viscous liquid of 99.5% purity. The organic concentrations of the prepared aqueous solutions ranged from about 0.1 to 2 M. Ternary solutions of the organic compounds, water and inorganic salt (sodium chloride (NaCl, 99.5%) or sodium sulfate ($Na_2SO_4$, 99.0%)), and an aqueous $Na_2SO_4$ solution were also prepared. Salts were added to yield inorganic concentrations between 1 to 2 M, and different mixing ratios with respect to the organics. All samples were well-mixed before use, yielding homogeneous aqueous solutions. Details of the studied compounds are given in Table 1, and of the samples in Table 2.

**Table 1.** Properties of the studied organic and inorganic compounds, including organic hydrates: molecular formula, aqueous solubility ($C_{sol}$ [g/100g]) at 25 °C, and equilibrium constant ($K$) for each hydration step.

| Formula | $C_{sol}$ [g/100g] | $K$ |
|---|---|---|
| CHOCHO | at least $67^a$ | $K_{hyd1} = 350^b$ |
| | | $K_{hyd2} = 207^c$ |
| $CH_3C(O)CHO$ | at least $67^a$ | $K_{hyd1} = 1279^c$ |
| $C_3H_8O_3$ | $\infty^d$ | - |
| NaCl | $100^e$ | - |
| $Na_2SO_4$ | $28.1^e$ | - |

$^a$ Saxena and Hildemann (1996). $^b$ Ervens and Volkamer (2010).
$^c$ Montoya and Mellado (1994). $^d$ Perry et al. (1997). $^e$ Haynes (2014).



**Table 2.** Sample compositions, with aqueous concentrations of each solute.

| Organic | Conc. [$\mathrm{mol\,dm^{-3}}$] | Inorganic | Conc. [$\mathrm{mol\,dm^{-3}}$] |
|---|---|---|---|
| | 1.94 | - | - |
| | 0.963 | - | - |
| | 0.495 | - | - |
| | 0.215 | - | - |
| CHOCHO | 0.0967 | - | - |
| | 1.62 | $\mathrm{Na_2SO_4}$ | 1.61 |
| | 0.458 | $\mathrm{Na_2SO_4}$ | 1.83 |
| | 1.93 | NaCl | 1.93 |
| | 0.501 | NaCl | 1.98 |
| | 0.976 | - | - |
| | 0.861 | - | - |
| | 0.494 | - | - |
| $\mathrm{CH_3C(O)CHO}$ | 0.497 | - | - |
| | 0.977 | $\mathrm{Na_2SO_4}$ | 0.979 |
| | 0.978 | NaCl | 0.980 |
| | 1.99 | - | - |
| $\mathrm{C_3H_8O_3}$ | 1.49 | $\mathrm{Na_2SO_4}$ | 1.49 |
| | 1.99 | NaCl | 1.99 |
| - | - | $\mathrm{Na_2SO_4}$ | 1.81 |

## 2.2 Liquid cell experiments

XAS measurements of liquid samples were performed at the BL3U beamline (Hatsui et al., 2004) of the UVSOR-III Synchrotron Facility, Okazaki, Japan. The 750 $\mathrm{MeV}$ storage ring was operated in a top-up mode with 300 $\mathrm{mA}$ current. The end station with the liquid flow cell system connected to the beamline has been described extensively by Nagasaka et al. (2018a).

Briefly, the liquid cell consists of two $\mathrm{Si_3N_4}$ membranes (area: $2 \times 2$ $\mathrm{mm^2}$, thickness: 100 $\mathrm{nm}$) in a chamber which is separated from the soft X-ray beamline under ultrahigh vacuum conditions ($< 10^{-5}$ $\mathrm{Pa}$) by using one $\mathrm{Si_3N_4}$ membrane ($0.2 \times 0.2$ $\mathrm{mm^2}$). The chamber is filled with flowing helium (He) at atmospheric pressure which can be varied by using outlet valve of He flow line. The liquid sample thickness was optimized for better transmission signal by changing the helium flow rate, and, therefore, the pressure exerted on the walls of the cell. The liquid sample in the cell can be exchanged using a pumping system

(Cole-Parmer Masterflex L/S) with adjustable flow rate. Here, the samples were pumped in the liquid cell with the flow rate of 5 $\mathrm{mL\,min^{-1}}$.





XAS spectra in transmission mode can be obtained based on the Beer-Lambert formula: $I = I_0 e^{-\mu x}$, where $I_0$ is the intensity of the incident radiation, $I$ is the transmitted intensity and $x$ is the thickness of the sample. The absorption intensity depends on the material linear attenuation coefficient $\mu$ (Stöhr, 1992). During each measurement the incident radiation was monitored

with a gold mesh placed before the liquid cell, so that the flux variations (<1%) due to the top-up mode were removed. The transmitted radiation $I$ is detected by a photodiode (IRD AXUV100) placed right after the liquid cell. The transmission signal was also measured without the sample, using a blank cell, to normalize the absorption of the membranes and possible contamination. The thickness of the liquid layer ($x$) was not precisely estimated. In order to avoid additional uncertainty on our results, the intensities of the XA spectra are given in arbitrary units. Here, we concentrate on the shape of the spectrum and

the energies of resonances rather than absolute absorption coefficients.

Carbon and oxygen K-edge (1s core level) absorption spectra are recorded by scanning photon energies between 282–305 eV and 525–550 eV, respectively. The energy range can be changed from the control system of the beamline, by changing the angle of a varied-line-spacing plane grating monochromator. Photon energy resolutions were approximately 0.2 eV for C and 0.4 eV for O K-edges, respectively. Reproducibility of the recorded spectra was verified by recording a few low statistics cases

several times. No significant changes were observed in the peak positions and the intensities of our spectra between the scans. During the experiment, we regularly recorded carbon and oxygen K-edge absorption spectra of pure water. The spectra were used for inspection of possible contamination in the system and as a reference for the characterization of the spectra of the aqueous solutions.

## 2.3    Data treatment

The C K-edge spectra of the samples were subtracted by the C K-edge spectra of pure water to remove any undesired absorptions arising from e.g. the absorption of water in the solution, that of $Si_3N_4$ membranes and that of carbon contamination in the beamline optics.

For the calibration of the C K-edge XAS spectra, we used XAS spectra of polymer films (proLINE) (Nagasaka et al., 2018a) scanned for energies from 284.5 eV to 288.5 eV with steps of 0.02 eV and 0.05 eV. After interpolation of the energy values

with step of the data points approximately 0.05 eV, the treated data were calibrated by the absorption peak of the polymer films at 285.07 eV (Nagasaka et al., 2018a).

In the C K-edge spectra of glyoxal, there were no pre-edge absorption peaks and the ionization edge was identified by fitting a sigmoid in the step of the spectra and by identifying the mean energy value from the inflection point. For the fitting of the spectra of methylglyoxal, we used the SPANCF curve fitting macro package (Kukk et al., 2001, 2005) which gives the energy

and intensity of the absorption peaks. The shape of the curves was an asymmetric Voigt shape and the ionization thresholds were modeled by an arctangent function.

The oxygen signal of the O K-edge spectra is overwhelmed by the signal from the water. Therefore, the signal of water was not subtracted from the O K-edge spectra, as in the case of the C K-edge spectra. The data were calibrated by the first peak of the polymer films at 530.88 eV (Nagasaka et al., 2018a). For comparison, we also measured the oxygen K-edge of glycerol

($C_3H_8O_3$) in pure water and aqueous mixtures with the inorganic salts, as it contains similar functional groups as the expected





hydrated forms of glyoxal and methylglyoxal (three hydroxyl groups bonded to three carbons and single C-H bonds), while excluding the possibility for reversible water addition or removal.

## 2.4 Quantum chemical calculations

The core ionization and excitation energies were evaluated within the $\Delta$SCF method by using the GSCF3 code (Kosugi and
Kuroda, 1980). Geometries of free glyoxal, methylglyoxal, hydroxides, and enol forms were optimized by ground-state SCF calculations within the second order Møller-Plesset perturbation theory using 6-311G* basis set (Frisch et al., 2004). According to our established method (Kosugi et al., 1992), core ionized states of these molecules were obtained by SCF calculations of their single core hole states with an extended basis set plus polarization basis functions (31111/21/3112/1*) for carbon and oxygen atoms and a double zeta basis set (42) for hydrogen (Huzinaga et al., 1984), and then by freezing the resultant core
hole orbitals, singlet core excited states of these molecules were obtained by SCF calculations of their single core-to-valence excitations. No diffuse functions to describe Rydberg states are included assuming there is no strong Rydberg-valence mixing and Rydberg states are negligible.

## 3 Results and discussion

Here we present results in terms of recorded XAS spectra for each sample of glyoxal, methylglyoxal and their mixtures with
inorganic salts. The carbon K-edge spectra are presented together with calculations for unhydrated and hydrated forms of glyoxal and methylglyoxal for better interpretation of the spectra. We also present oxygen K-edge spectra of aqueous solutions with glycerol and their ternary solutions containing inorganic salts.

### 3.1 Carbon K-edge

#### 3.1.1 Quantum chemical calculations

We calculated ionization potentials (IP) and C 1s-$\pi$* bound–state excitation energies for unhydrated and hydrated species of glyoxal and methylglyoxal in gas phase. Quantum chemical calculations are extremely difficult to be accurate for complex systems (e.g. liquids) due to computational limitations. Gas phase calculations can be used instead to describe the electronic excitations. Typically, the core-to-valence excitation energies of molecules in gas phase show only small shifts compared to the excitation energies of these molecules in liquid phase or as a part of a cluster. However, the IP of the molecule is higher
in the gas phase compared to condensed phases. The Rydberg series converging to the ionization threshold are shifted in higher energies for the lower Rydberg and shifted to lower energies or quenched for the higher Rydberg by the surroundings (Flesch et al., 2004; Nagasaka et al., 2018b). The results of the quantum chemical calculations are presented in Table 3. In the case of glyoxal, we observe that the excitation energies of C=O bonds of the unhydrated and monohydrated form differ by approximately 0.5 eV. The IP of the C=O sites (295.4 for the unhydrated form and 295.15 eV for the monohydrated form) are
higher than these of the C-OH sites by 0.54 and 0.29 eV, respectively.





**Table 3.** Calculated C 1s ionization potentials (I.P.) and C 1s-$\pi$* bound-state excitation energies (E$\pi$*) below I.P. in eV with oscillator strengths (f) of sites present in hydrated forms of glyoxal (gly) and methylglyoxal (meg), shown in Figs. 1 and 2. For sites with more than one carbon atom, we denote with bold (**C**) the carbon atom that is related to the calculated value.

| Compound | Site | IP | E$\pi$* | f |
|---|---|---|---|---|
| gly | C=O | 295.40 | 286.18 | 0.046 |
| gly hydrated | C=O | 295.15 | 286.66 | 0.065 |
| | C-OH | 294.86 | 289.97 | 0.001 |
| gly dihydrated | C-OH | 294.86 | 289.99 | 0.001 |
| meg | C=O | 295.07 | 286.23 | 0.049 |
| | **C**=O(CH$_3$) | 294.96 | 286.51 | 0.047 |
| | CH$_3$ | 291.82 | 286.60 | 0.000 |
| meg hydrated | C-OH | 294.53 | 290.27 | 0.001 |
| | **C**=O(CH$_3$) | 294.73 | 287.05 | 0.070 |
| | CH$_3$ | 291.90 | 287.64 | 0.000 |
| meg dihydrated | C-OH | 294.42 | - | - |
| | **C**-OH(CH$_3$) | 294.30 | - | - |
| | CH$_3$ | 291.31 | - | - |
| enol form of meg | C=O | 294.89 | 286.87 | 0.054 |
| | **C**-OH(CH$_2$) | 293.13 | 286.90 | 0.015 |
| | CH$_2$ | 290.85 | 284.85 | 0.017 |
| enol form of meg hydrated | C-OH | 294.25 | 291.96 | 0.002 |
| | **C**-OH(CH$_2$) | 292.87 | 287.53 | 0.038 |
| | CH$_2$ | 290.58 | 285.91 | 0.019 |

For methylglyoxal, the C 1s-$\pi$* excitation energies were calculated for the unhydrated, monohydrated and dihydrated forms. We also calculated energies of enol forms of methylglyoxal and hydrated methylglyoxal because they have been identified as possible structures in aqueous methylglyoxal solutions by Krizner et al. (2009). They applied DFT calculations to study the hydration of 1 M of methylglyoxal at 298 K and showed that formation of enol structures and aldol condensation products, 140 which have in their chemical structure a C=C bond, are the thermodynamically most stable species in the methylglyoxal system.





This is a very favorable chemical path when methylglyoxal is hydrated, and thus taken into account in our calculations as well. Figure 2 shows the transition from the monohydrated methylglyoxal to the enol formation which is the intermediate step for the aldol condensation products.

The excitation energies of C=O sites at 286.23–287.05 eV were lower than those of C-OH sites of C=C formed by keto-enol

tautomerism of the monohydrated methylglyoxal, but higher than the excitation energies of the CH$_2$ sites of C=C bonds, at 284.85–285.91 eV. The IP of the C=O sites (294.73–295.07 eV) are higher by approximately 0.7 eV than the C-OH and C-OH(CH$_3$) sites (294.25–294.53 eV) but much higher than the C-OH(CH$_2$) sites of the enol structure at 293.13 eV and 292.87 eV. This indicates that the hydration and keto-enol tautomerism lower the C 1s ionization threshold.

**Figure 2.** Enol formation from hydrated form of methylglyoxal.

In Table 3 we also present the calculated oscillator strengths ($f$).

**3.1.2 Glyoxal**

Figure 3 presents the recorded spectra for the C 1s absorption edge of binary glyoxal–water and ternary solutions formed by addition of either NaCl or Na$_2$SO$_4$ together with the excitation energies of our quantum chemical calculations. The results of the quantum calculations are presented with colored lines that correspond to hydrated and unhydrated molecules in gas phase, and the length of the lines is scaled in respect with the calculated oscillator strengths. Apart from the absorption probability of

a molecule, the absorption intensity of the presented spectra is also related to the relative amount of the species in the solution and the interactions that take place in the liquid phase, and thus we cannot directly use oscillator strengths as expected intensity ratios in the experimental spectra.

The spectra with and without addition of inorganic salts have only a single feature at around 289.6 eV. The absorption edges of the recorded spectra are presented in Table 4. The absorption energies of different solutions differ by 0.1–0.2 eV,

which is just within the photon energy resolution of the measured spectra. We observe that there is a trend to increase the absorption edge energy upon addition of salt or decrease in concentration. Based on the calculated energies (Table 3), this energy is close to the C-OH excitation energies of the monohydrated and dihydrated form, calculated at 289.97 eV and 289.99 eV, respectively. Photon energies above 291 eV exceeds the C 1s binding energy and ionization starts. We do not observe any peaks at the calculated C=O energies of the unhydrated and monohydrated glyoxal at 286.18 eV and 286.66 eV, respectively.

This indicates that the dihydrated form of glyoxal overwhelmingly dominates in our liquid solution samples.



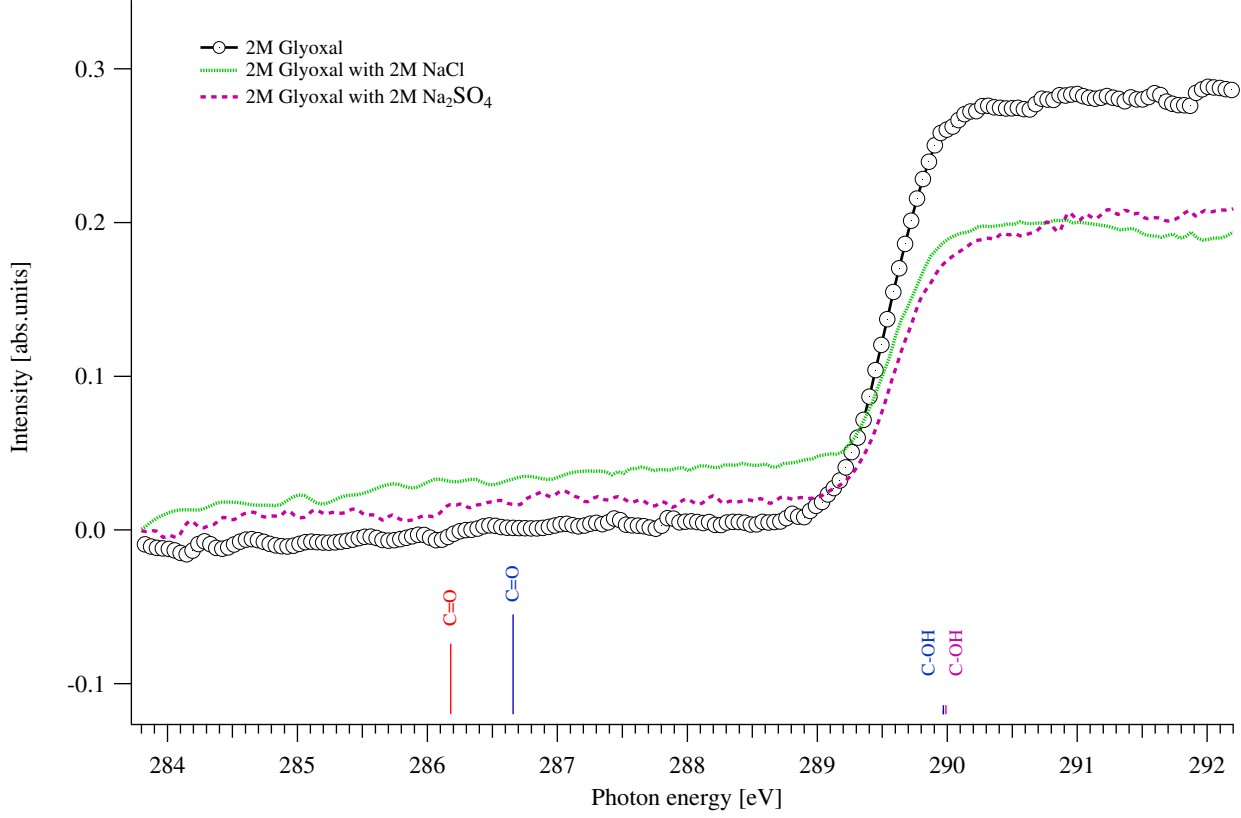

**Figure 3.** C K-edge spectra of glyoxal–water (2 M) binary solution and 1:1 molar mixtures with NaCl and Na$_2$SO$_4$. The lines present the calculated C 1s $\rightarrow \pi^*$ excitation energies. The color of the lines corresponds to different hydrated species: red: unhydrated glyoxal, blue: monohydrated glyoxal, purple: dihydrated glyoxal. The lines have been scaled according to the calculated oscillator strengths.

The dominance of hydrated species in aqueous glyoxal has been reported before. Measurements of near-UV molar absorptivities of aqueous solutions at 25 °C have shown that 98% of glyoxal is in dihydrated form and the rest is in monohydrated form (Malik and Joens, 2000). Kua et al. (2008) applied density functional theory (DFT) calculations to find the favorable hydration paths of glyoxal and of the subsequent products of hydration and used the Poisson-Boltzmann (PB) continuum approximation

to describe the interaction with water. Their results show that the hydrated species are the most thermodynamically favorable and they can subsequently be involved in reactions to form oligomers. Yu et al. (2011) studied aqueous solutions with 1 M glyoxal with Nuclear Magnetic Resonance (NMR) spectroscopy. In agreement to our study, they did not detect unhydrated glyoxal and concluded that the glyoxal in the solution is mainly in dihydrated form and contains oligomers. Kua et al. (2008) and Yu et al. (2011) identified potential oligomers as hydration products of aqueous glyoxal. Therefore, oligomers of glyoxal

that contain C-OH bonds may be present in our solutions.





**Table 4.** Absorption edge of aqueous solutions of pure glyoxal and their mixtures with inorganic salts. Error bars are estimated to be $\pm$ 0.1 eV.

| Aqueous solution | Absorption edge [eV] |
|---|---|
| 0.5 M $C_2H_2O_2$ | 289.6 |
| 0.5 M $C_2H_2O_2$ + 2 M $Na_2SO_4$ | 289.7 |
| 0.5 M $C_2H_2O_2$ + 2 M NaCl | 289.7 |
| 1 M $C_2H_2O_2$ | 289.5 |
| 2 M $C_2H_2O_2$ | 289.5 |
| 2 M $C_2H_2O_2$ + 2 M $Na_2SO_4$ | 289.6 |
| 2 M $C_2H_2O_2$ + 2 M NaCl | 289.6 |

### 3.1.3 Methylglyoxal

In Fig. 4 we present the spectra of C 1s of methylglyoxal in aqueous mixtures. We observe three distinct absorption features at 285.8, 287.9 and approximately 290 eV. Figure 5 shows five peaks, denoted as A, B, $C_1$, $C_2$ and $C_3$, that were identified during the fitting process of the spectum of methylglyoxal in aqueous phase. We correlate these peaks with the absorption energies of functional groups presented in our solutions (Table 5). We also present the excitation energies and ionization potentials of our quantum chemical calculations.

**Peak A**

The first peak designated as A (Fig. 5) is found at 285.8 eV. We assign this peak to the C=C-OH moiety of the enol form of the monohydrated methylglyoxal, in good agreement with the predicted C=C excitation energy of 285.91 eV. Usually, the C=C peaks are found at 285 eV, but increase in the energy of C=C transitions has been observed before in organic compounds with similar structure. Phenols and quinones in solid phase were studied by Solomon et al. (2009), using Scanning Transmission X-ray Microscopy (STXM) under a He atmosphere. These compounds have the same characteristic moiety, and excitations between 286.05 and 286.35 eV were attributed to C 1s $\rightarrow \pi^*_{C=C}$. C=C bonds can also result from radiation damage of the sample. However, consecutive scans of the solutions did not show any significant changes in the spectra, and thus radiation damage is considered to be small in our experiments.

**Peak B**

The next strong peak in Fig. 4 at 287.9 eV (peak B in Fig. 5) can be assigned to the C-OH($CH_2$) transitions in the enol form of monohydrated methyglyoxal. The calculated energy of this transition is at 287.53 eV (lower by 0.4 eV).

**Peaks $C_1$–$C_3$**





The asymmetrical shape of our spectra suggests underlying peaks. We fit three peaks, according to the observed shoulders on the two main peaks, at 286.5 eV (peak $C_1$), 287.1 eV (peak $C_2$) and 288.6 eV (peak $C_3$). The predicted C=O absorption energies are at 286.51 eV and 287.05 eV for the unhydrated and monohydrated methylglyoxal in gas phase, respectively. Latham et al. (2017) studied solid organic compounds in ultrahigh vacuum conditions (UHV), applying near-edge X-ray absorption fine structure (NEXAFS) and has also assigned the C=O transitions at 286.5 eV. Russell et al. (2002) applied

soft X-ray spectromicroscopy at atmospheric pressure and Tivanski et al. (2007) used STXM with NEXAFS at 0.5 atm to characterize functional groups of dry particles and they both correlated the C=O transitions with absorption energies at around 286.7 eV. The calculated and reported values for C=O transitions are lower than the respective peak at 288.3 eV in the gas phase spectrum (presented in Fig. S1 of the Supporting Information). The calculated C 1s-$\pi$* excitation energies of the C-H sites, are at 286.6 and 287.64 eV for the unhydrated and monohydrated form, respectively. C-H Rydberg states have not

been calculated but could be mixed with the calculated states and could be interacted with surrounding water molecules in aqueous solutions. C-H transitions have been assigned by Latham et al. (2017) at 287.3 eV and at the energy range from 287 to 288.4 eV by Russell et al. (2002). In conclusion, in this energy range we expect peaks related to C=O excitations associated to the unhydrated methylglyoxal, C=O(CH$_3$) excitations of the monohydrated methylglyoxal and also C-H excitations of the hydrated species. Based on previous studies, we expect to have mainly monohydrated and dihydrated forms of the compound

and a negligible fraction of unhydrated methylglyoxal (Nemet et al., 2004; Krizner et al., 2009). C=O and C-H excitations from hydrated species that contain enol structures and originate from monohydrated methylglyoxal should also be considered as they can co-exist in the solutions. Due to the variety of possible species containing C=O and C-H bonds in the aqueous solutions, we suggest an energy range from 286.5 to 288.6 eV for the transitions of C=O and C-H groups.

**Feature at 290 eV**

The wide feature at around 290 eV in Figs. 4 and 5 is related to C-OH transitions and ionization of C 1s electrons. The calculated C-OH absorption energy of the monohydrated form at 290.27 eV matches well with the energy from where the broad structure starts. The IP are expected to be shifted by some eVs compared to the calculated, depending on the surroundings where nearest neighbor is dominant.

        In the case of methylglyoxal, addition of salts did not introduce observable shifts to absorption energies. Due to the presence

of overlapping peaks in the spectra of the salt-free systems, we cannot estimate the absolute concentrations of unhydrated and hydrated methylglyoxal and thus, we cannot observe how the addition of salts affect the concentration of the hydrates. Changes in the relative ratios of unhydrated and hydrated forms would be expected due to changes in the molecular interactions (water–methylglyoxal, water–water) when ions co-exist in the solution. Such phenomena have been reported in previous experimental (Waxman et al., 2015) and computational (Toivola et al., 2017) studies, which have shown decrease of the co-

solubility of methylglyoxal in aqueous solutions with addition of NaCl and Na$_2$SO$_4$ (salting out effect). Here we observe a small change in the relative intensities between the peaks A and B in pure methylglyoxal solution compared to solution spiked with Na$_2$SO$_4$, but the shape of the background also changed in the spectrum and thus these changes remain inconclusive.





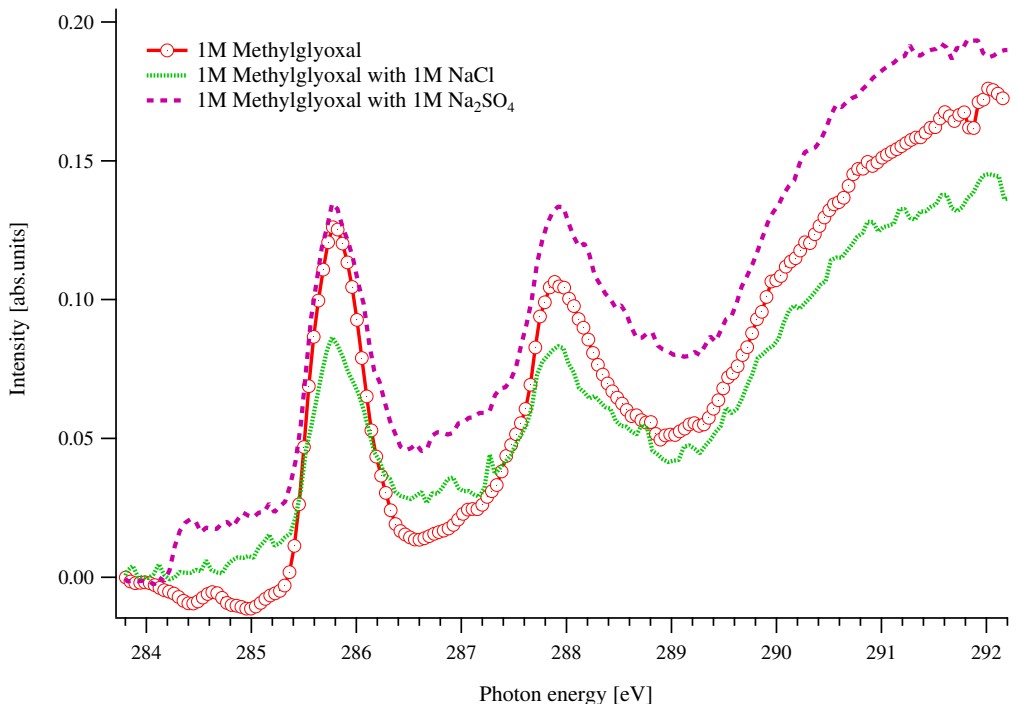

**Figure 4.** C K-edge spectra of 1 M aqueous solution of methylglyoxal together with ternary mixtures with 1 M of NaCl and 1 M of $Na_2SO_4$.

**Table 5.** Summary of the identified peaks of carbon K-edge and the absorption energies for methylglyoxal in this study.

| Peak | Energy [eV] | Functional group | Assignment |
|------|-------------|------------------|------------|
| 1 M $CH_3C(O)CHO$ | | | |
| A | 285.8 | $C=C_{(enol)}$ | $1s \rightarrow \pi^*$ |
| B | 287.9 | $C-OH(CH_2)$ | $1s \rightarrow \pi^*$ |
| $C_1$-$C_3$ | 286.5-288.6 | C=O and C-H | |

## 3.2 Oxygen K-edge

Figures 6, 7 and 8 show the O K-edge absorption spectra of organic-water binary solutions and comparison of these when

$Na_2SO_4$ or NaCl is added. The spectra have been normalized to the background preceding the water pre-peak. The two

structures can be characterized as a pre-peak and a main peak.

As mentioned in Sect. 2.3, in the case of oxygen, the water background has not been subtracted. Due to the abundance of

water in the solutions, the spectra are similar to that of water. The pre-edge peak at 535 eV is related to the transition to an





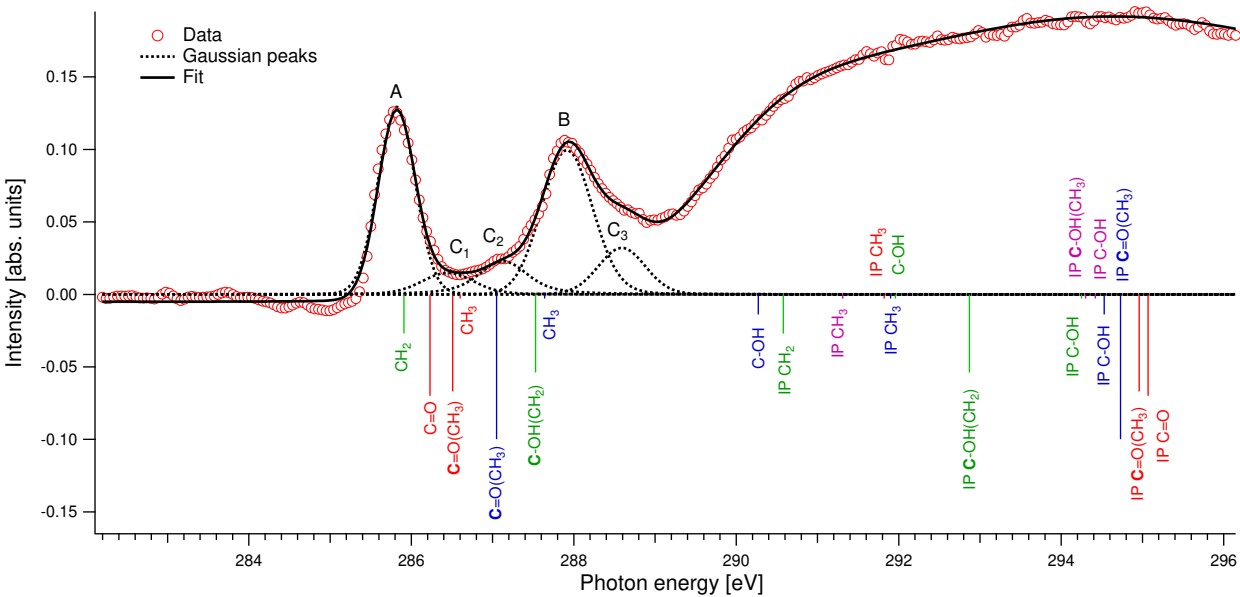

**Figure 5.** Normalized spectrum of 1 M methylglyoxal with the identified and calculated peaks and ionization steps. The calculated values are presented with red color for the unhydrated methylglyoxal, blue color for the monohydrated methylglyoxal, purple color for the dihydrated methylglyoxal and green color for the enol form. The lines have been scaled according to the calculated oscillator strengths.

orbital with the $4a_1$ character in a water molecule, and the main edge is related to the transition to the continuum threshold

of bulk water (Sellberg et al., 2014; Pylkkänen et al., 2010). Changes of the pre-edge peak can be related to changes in the hydrogen bond network (Nilsson et al., 2010; Velasco-Velez et al., 2014; Bluhm et al., 2002). We observe that for all the organic compounds (Figs. 6, 7 and 8), the ratio of the pre-peak feature to main peak is smaller than in water. We do not observe any relative changes in spectral features as a function of concentration of the organic compound. In ternary solutions with $Na_2SO_4$ the ratio of the main edge region (537-540 eV) to post edge (540-544 eV) differs from all the other solutions. This has been

previously described by Niskanen et al. (2015) to be related to tetrahedral geometry of the system. In solutions with $Na_2SO_4$ we also observe that the pre-edge peaks are blue-shifted similarly to the $Na_2SO_4$ aqueous solution. The energy difference between the pre-edge peaks of pure water and the ternary solutions with $Na_2SO_4$ is approximately 0.1 eV for the solutions of glyoxal containing 2 M of $Na_2SO_4$ and 0.05 eV for the solution of methylglyoxal containing 1 M of $Na_2SO_4$. This shift is plausible considering that $Na^+$ ions can have strong interactions with oxygen of water (Nagasaka et al., 2015, 2017) and that

$SO_4^{2-}$ tends to form hydrogen bonds with water molecules (Niskanen et al., 2015).

Figure 8 presents O K-edge spectra of 2 M aqueous glycerol and its mixture with NaCl and $Na_2SO_4$. Similarly to glyoxal and methylglyoxal spectra, the shape of the pure solution and the one spiked with NaCl are very close to pure water. Again, when $Na_2SO_4$ is added, the prepeak shifts to higher energies (by 0.1 eV) and the ratio of the main and post edge features changes, thus getting closer to the shape of pure $Na_2SO_4$ solution.





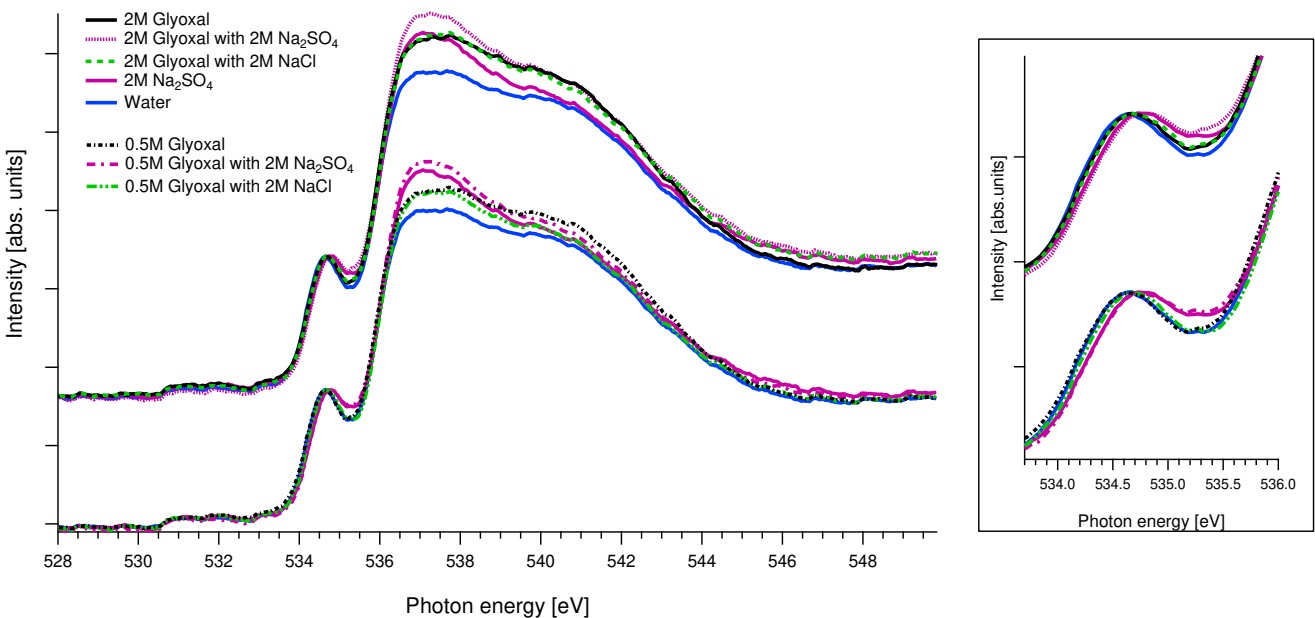

**Figure 6.** Left: O K-edge spectra of aqueous solutions of glyoxal together with ternary mixtures with 2 M of $Na_2SO_4$ and 2 M of NaCl. For comparison we present also the O K-edge spectrum of pure water and of aqueous solution with 2 M of $Na_2SO_4$. Right: Zoom on the pre-peak of the spectrum.

In conclusion, O K-edge spectra were found to be sensitive neither to the organic component at studied concentrations nor to addition of NaCl. However, addition of $Na_2SO_4$ affected the spectra as they started to resemble more pure $Na_2SO_4$ solution without any organics. Thus, no $Na_2SO_4$–organic interaction can be confirmed using this method.

### 3.3 Atmospheric implications

Both glyoxal and methylglyoxal are volatile organic compounds (Kielhorn et al., 2004; Liggio et al., 2005; Kalberer et al., 2004) and thus expected to be significantly present in the gas phase in the atmosphere. Nevertheless, both contribute significantly to atmospheric SOA (Volkamer et al., 2006; Fu et al., 2008). Here, we confirm experimentally that glyoxal, to the sensitivity of our measurements, exists entirely in its fully hydrated form in aqueous solution, in agreement with previous studies (Yu et al., 2011; Malik and Joens, 2000; Kua et al., 2008). Our study shows that hydrated methylglyoxal furthermore takes part in enol formation and aldol condensation reactions which contributes further to shifting the gas-particle phase partitioning towards the aqueous particle phase. Both reactions can therefore lead to significant SOA formation.

The presence of organic solute, such as glyoxal or methylglyoxal, in the aqueous solution decreases the equilibrium water vapor pressure $p$ over the solution compared to the equilibrium vapor pressure of a water solution (solute effect). According to Köhler theory (Köhler, 1936), increased uptake of SOA to the aqueous phase promotes hygroscopic growth, as well as





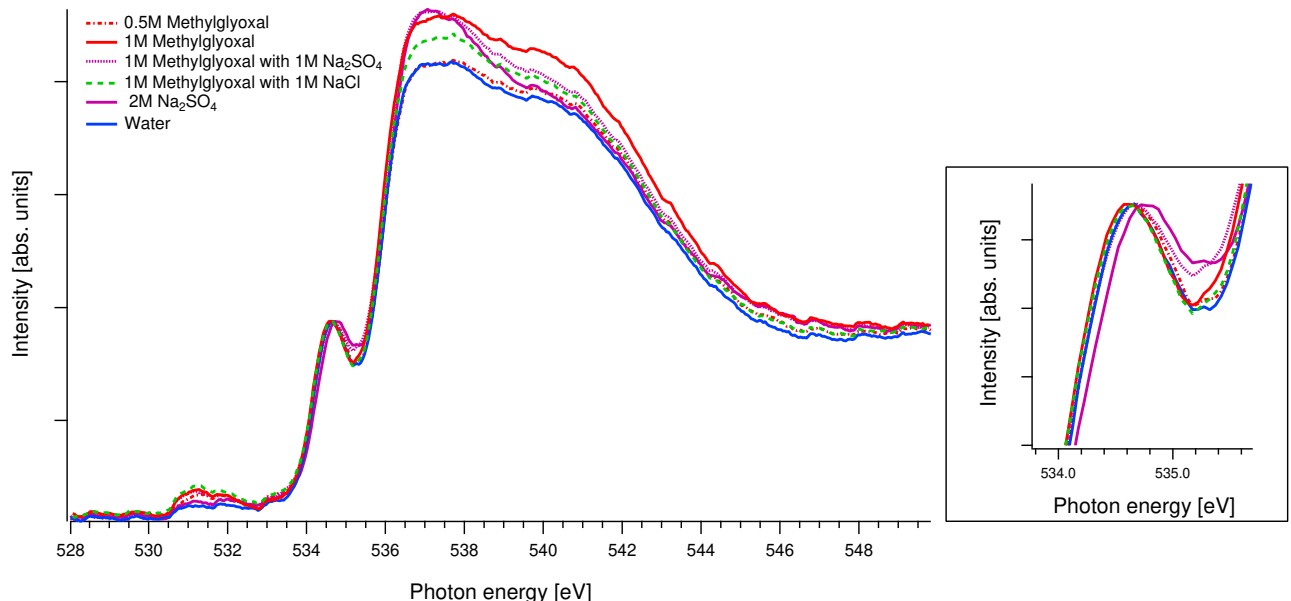

**Figure 7.** Left: O K-edge spectra of 1 M aqueous solution of methylglyoxal and aqueous solutions of methylglyoxal with 1 M of $Na_2SO_4$ and 1 M of NaCl. We also present the recorded spectra of pure water and binary water-2 M $Na_2SO_4$ solution. Right: Zoom on the pre-peak of the spectrum.

atmospheric cloud formation by lowering the humidity threshold for droplet activation (e.g., Prisle et al., 2010; Hansen et al., 265    2015).

## 4    Conclusions

We applied carbon and oxygen K-edge XAS using synchrotron radiation to study aqueous solutions of atmospheric highly oxygenated organics, glyoxal and methylglyoxal. We recorded absorption spectra for the carbon and oxygen K-edges and we measured as reference also oxygen K-edge spectrum of aqueous glycerol.

Glyoxal was found to be fully hydrated in aqueous solutions, as no peaks related to transitions in C=O group were identified in our spectra. C 1s XAS of methylglyoxal had two well-defined peaks before the broad absorption continuum, and were assigned using quantum chemical calculations to C=C and C-OH(CH$_2$) moieties of the enol form of monohydrated methylglyoxal. Transitions in C=O and C-H functional groups contributed to the C 1s XAS as broad shoulders in the energy range from 286.5–288.6 eV. They are found in the monohydrated and dihydrated forms of the compound and also from products of 275    keto–enol tautomerism followed by aldol condensation reactions.





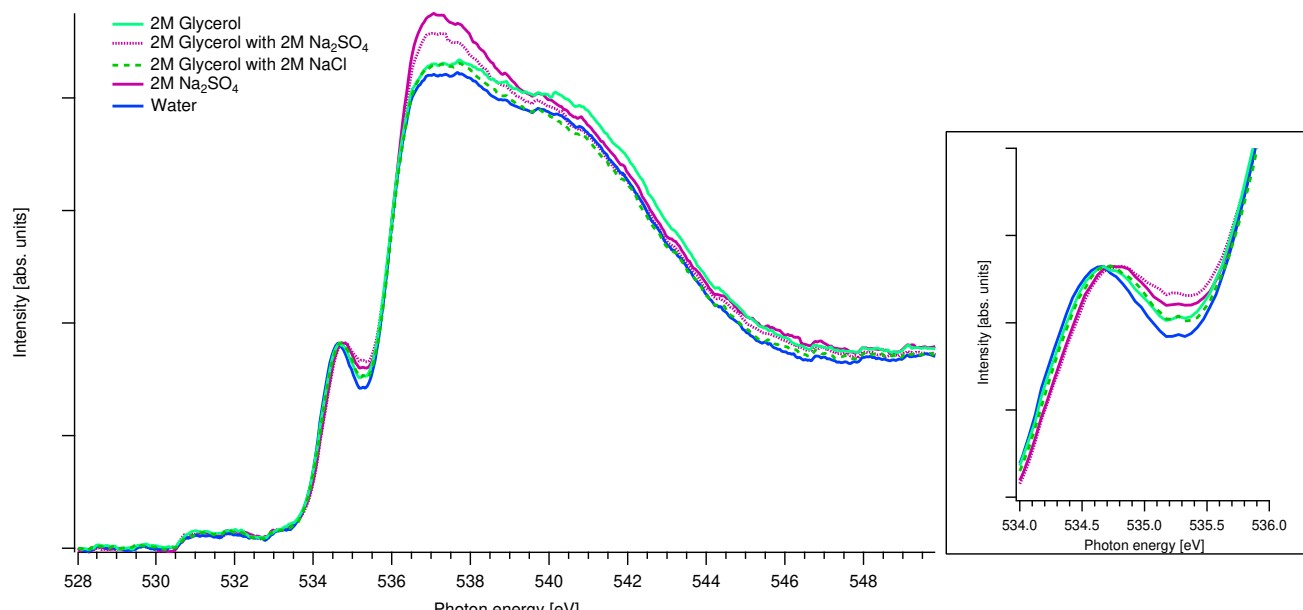

**Figure 8.** Left: Recorded spectra of aqueous solutions of glycerol, in oxygen K-edge. Comparison of aqueous solution of 2 M of glycerol and 1:1 molar mixtures with $Na_2SO_4$ and NaCl. The spectra of binary water–2 M $Na_2SO_4$ solution and pure water are also presented for comparison reasons. Right: Zoom on the pre-peak of the spectrum.

The oxygen K-edge spectra were similar to water, due to low solute concentrations and we only observed small changes in the case of mixtures with $Na_2SO_4$: a small shift of the pre-peak to higher absorption energies and increase of intensity of the first feature after the main absorption edge at 538 eV due to strong interactions of $Na^+$ and $SO_4^{2-}$ ions with the atoms of water.

In both carbon and oxygen K-edge spectra, we did not observe any effects of salting in/out interactions on the shape of the
spectra. This indicates that XAS might not be sensitive enough to see additional changes to the hydrate formation equilibrium, as e.g. demonstrated by Kurtén et al. (2014). As a rough estimate of the sensitivity, even at the highest 2 M concentration, we were not able to identify any unhydrated form of glyoxal, even if 2% of it has been reported before (Malik and Joens, 2000). However, our finding that glyoxal exists entirely on covalently bond dihydrate form in aqueous solution is in line with the results of Yu et al. (2011).

Our study presents the first experimental verification that methylglyoxal in water solution contains not only monohydrated and dihydrated forms of the compound but also other chemical species with enol structures. The results contribute to explain the relatively high contribution of glyoxal and methylglyoxal to atmospheric SOA, despite their high saturation vapor pressures.





*Author contributions.* NLP conceived, planned and supervised the project and secured beamtime and funding. GM, HY, MP, NLP and MH carried out the experiments on liquid samples. MN performed the gas-phase measurements of glyoxal and methylglyoxal. NK performed the
quantum chemical calculations. GM analyzed the experimental data with supervision and assistance of MP. All the authors contributed to discussions on the results. GM wrote the paper with assistance of JJL and contributions from all co-authors.

*Competing interests.* The authors declare that they have no conflict of interest.

*Acknowledgements.* We thank the staff of BL3U beamline of the UVSOR-III for their support in conducting experiments. This project has received funding from the European Research Council (ERC) under the European Union's Horizon 2020 research and innovation programme
(grant agreement No 717022). N.L.P., J.J.L., M.P. and M.H. gratefully acknowledge funding from the Academy of Finland (grant No. 308238, 314175 and 296338). This project has received funding from the European Union's Horizon 2020 research and innovation programme under the Marie Skłodowska–Curie (grant agreement No. 713606).





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
