# Peer review of "Aqueous phase behavior of glyoxal and methylglyoxal observed with carbon and oxygen K-edge X-ray absorption spectroscopy."

_Atmospheric Chemistry and Physics, 2020_

## Referee Comment (RC1) · Anonymous Referee #3 · 26 Aug 2020

This work describes the use of X-ray absorption spectroscopy to study solutions of glyoxal and methylglyoxal mixed with NaCl or Na2SO4, in order to simulate aqueous aerosol environments. While the method turned out to be too insensitive to observe the effects of organic – inorganic interactions (between, for example, glyoxal and sulfate), which seem to have been the original goal of the study, it did produce clear observations of enol forms of methylglyoxal in solution, the main new result. Glyoxal's tendency to be fully hydrated was also observed in the lack of any C=O signal, consistent with the results of several other studies using different spectroscopic methods. Together, these observations confirm the ability of glyoxal and methylglyoxal to hydrate and react in other ways when dissolved in water, and therefore form aqueous SOA in the

atmosphere.

Specific comments:

Line 7: The abstract suggests that the X-ray spectra of methylglyoxal imply the presence of the dihydrate form, but there is little evidence of this form according to Figure 5. Any dihydrate transitions appear to be buried in the ionization peak. I think the data does not imply the dihydrate's presence or absence.

Line 10: Since organic - inorganic interactions have been observed for these species by other methods, a statement about a lack of method sensitivity would be more appropriate than one about the weakness of these interactions. They are evidently strong enough to greatly increase measured Henry's law coefficients. (Kampf, Waxman et al. 2013)

Figure 3: Are the decreases in the C-OH ionization peak sizes caused by salt addition evidence of a chemical change? If so, this would be the opposite of what was observed for Na2SO4 by Yu et al. (Yu, Bayer et al. 2011). Could the authors address this?

Figure 4: Are the decreases in the size of the A and B peaks caused by salt addition (especially for NaCl) evidence of a shift away from enol forms?

Line 274: This sentence states that C=O and C-H functional groups are found in the dihydrate form of methylglyoxal. For C=O this statement is false.

Technical Corrections:

Table 3: Why are excitation energies not listed for dehydrated methylglyoxal? The text states that they were calculated in line 136.

Line 159: Are these differences of 0.1 – 0.2 eV significant, given that they are "within the photon energy resolution"? Or are they just random variation? If random, Table 4 should be moved to the SI section or eliminated from discussion.

In Table 5, the functional group entry for peak B should also be labelled "enol", if I

[Figure]

understand Figure 5 correctly. Also, no oscillator appears to line up with the C3 peak, so the origin of its assignment in Table 5 is unclear.

Figures 6-8 have two deep pink lines in the legend that look identical, but not in the figure itself.

Line 276: This sentence calls 1 – 2 M "low solute concentrations," which is somewhat misleading. It seems like the issue is rather the large signals of water.

Cited references:

Kampf, C. J., E. M. Waxman, J. G. Slowik, J. Dommen, L. Pfaffenberger, A. P. Praplan, A. S. H. Prévôt, U. Baltensperger, T. Hoffmann and R. Volkamer (2013). "Effective Henry's Law Partitioning and the Salting Constant of Glyoxal in Aerosols Containing Sulfate." Environmental Science & Technology 47(9): 4236-4244.

Yu, G., A. R. Bayer, M. M. Galloway, K. J. Korshavn, C. G. Fry and F. N. Keutsch (2011). "Glyoxal in aqueous ammonium sulfate solutions: products, kinetics, and hydration effects." Environmental Science and Technology 45: 6336-6342.

---

## Referee Comment (RC2) · Anonymous Referee #4 · 1 Nov 2020

The authors use XAS to study the structure of glyoxal and methylglyoxal in aqueous solutions with varying levels of inorganic salts. Quantum chemical calculations (that include electron correlation effects) are used to estimate ionization and excitation energies for hypothesized molecules in the gas-phase with appropriate discussion of uncertainties in using them to interpret condensed-phase spectra. From the C K-edge spectra, glyoxal is found to be completely hydrated due to observation of COH and lack of C=O, while products of enol structures and aldol condensation products that include C=C bonds are observed for methylglyoxal (in addition to hydrated forms suggested by C=C and COH bonds). The O K-edge is found to be generally insensitive for examining the structure of these organic molecules in these systems. The manuscript

is clearly written and interpretations are sensible. The work is of interest to the Atmospheric Chemistry and Physics community and is recommended for publication with minor changes.

Minor comments:

In the Introduction, there is no mention of relevant literature with regards to characterizing molecular structure and electrolyte-nonelectrolyte interactions in relevant systems. This is actually partially covered later in the discussion, but the work should be placed in context of prior art up front.

These ionic strengths are quite low by atmospheric standards so there should be a caveat regarding the limitations of this study.

The authors introduce the "NEXAFS" later in the manuscript as if it were a different technique, but it's not clear that this is actually different from the technique used by the authors referred under broader term, "XAS".

Apart from studying water (as cited by the authors), the O K-edge has generally not been found to be useful for characterizing specific structures of organic compounds (apart from indicating total oxygen content) in past work, and also seems to be the case here. The O K-edge does not really make a contribution to the main findings of the manuscript and should probably be summarized in a few sentence and removed otherwise (or placed in supplemental).

Table 4 also seems superfluous and can be moved to the supplemental since it can be summed up concisely in the text.

Can the authors elaborate on how the measured absolute intensity or optical density calibration is used in the experiments?

The way the units is presented is not entirely clear since the authors talk about measured beam intensity and absorption intensity (abs. = absorbance units?). Can authors clarify in one of the Figure captions? **ACPD**
The inline equation at the top of pg. 5 should be the Lambert formula since Beer's contribution comes from introducing the molar or number concentration to the linear attenuation coefficient of the material.

The authors seem to use the term "salting in/out" to generally refer to nonideal electrolyte-nonelectrolyte interactions instead of the consequence of their interactions, unless they are expecting a specific type of change in the spectra that was not made explicit in the manuscript.

Fig. 3: One of the vertical lines appear red while the other appears purple. It should be specified that it's these vertical lines that are scaled according to oscillator strengths and not the spectra (which are also drawn as lines).

Fig. 5. Since the overall fit is shown, can the authors display all the curves that are fitted (not just the Gaussian peaks)?

The authors talk about "IP of the C=O sites" (and for other groups) but would maybe more naturally be referred to as the "IP of the C=O moiety" in this context?

"SCF" is not defined.

**ACPD**

---

## Author Comment (AC1) · 4 Dec 2020

**Aqueous phase behavior of glyoxal and methylglyoxal observed with carbon and oxygen K-edge X-ray absorption spectroscopy.**

Georgia Michailoudi, Jack J. Lin, Hayato Yuzawa, Masanari Nagasaka, Marko Huttula, Nobuhiro Kosugi, Theo Kurtén, Minna Patanen, and Nønne L. Prisle

We thank both reviewers for their work and constructive comments. Below we provide our responses to each comment in a point-by-point fashion. The reviewers' comments are reproduced in *italics*, our responses in **bold** and quotes from the revised manuscript in **red bold** font.

**Anonymous reviewer #3**

*Line 7: The abstract suggests that the X-ray spectra of methylglyoxal imply the presence of the dihydrate form, but there is little evidence of this form according to Figure 5. Any dihydrate transitions appear to be buried in the ionization peak. I think the data does not imply the dihydrate's presence or absence.*

**Authors' reply:**

**Thank you very much. Indeed the statement that dihydrated form of methylglyoxal is present in the solution cannot be observed in Fig. 5 and is based on the weak C=O signal of our spectra and the results of previous studies. We therefore rephrase the sentence "This implies ... monohydrates" (lines 7 and 8) to:**

**"The relatively low intensity of C=O transitions implies that the monohydrated form of methylglyoxal is not favored in the solutions. Instead, the spectral intensity is stronger in regions where products of aldol condensation and enol tautomers of the monohydrates contribute."**

*Line 10*: Since organic - inorganic interactions have been observed for these species by other methods, a statement about a lack of method sensitivity would be more appropriate than one about the weakness of these interactions. They are evidently strong enough to greatly increase measured Henry's law coefficients. (Kampf, Waxman et al. 2013 [2])

**Authors' reply:**

**Thank you very much for the suggestion. We rephrase the sentence: "indicating that the organic–inorganic interactions at the studied concentrations are not strong enough to affect the spectra in this work" (lines 9 and 10) to:**

**"indicating that the XAS in the near-edge region is not very sensitive to these intermolecular organic–inorganic interactions at the studied concentrations".**

*Figure 3: Are the decreases in the C-OH ionization peak sizes caused by salt addition evidence of a chemical change? If so, this would be the opposite of what was observed for $Na_2SO_4$ by Yu et al. (Yu, Bayer et al. 2011 [12]). Could the authors address this?*

**Authors' reply:**

**Thank you very much. As we explain in the paper, since we cannot precisely determine the thickness of the liquid layer, we cannot directly compare the intensities of the spectra between different solutions. The overall signal in C 1s region would actually be expected to increase upon addition of salt to the solutions due to ionization of close-lying L-shells of Cl and S. The cross-sections of these orbitals are significant compared to the C K-shell in this energy range [11]. However, a change in the hydration state would be seen as new spectral features as predicted by the calculations. The following text is added in line 165:**

**"According to the calculations, a change in hydration state upon addition of salts towards mono or dehydrated form of glyoxal would have shown up as new spectral features around 286–287 eV, which is not observed."**

*Figure 4: Are the decreases in the size of the A and B peaks caused by salt addition (especially for NaCl) evidence of a shift away from enol forms?*

**Authors' reply:**

**To clarify this, we modify the sentence "Here we observe a small change in the relative intensities between the peaks A and B in pure methylglyoxal solution compared to solution spiked with $Na_2SO_4$, but the shape of the background also changed in the spectrum and thus these changes remain inconclusive" (lines 225–227) to:**

**"Here we observe a small change in the relative intensities between the peaks A and B in pure methylglyoxal solutions, compared to solutions containing $Na_2SO_4$ and NaCl. In salt-containing solutions, the relative intensity of peak A ($CH_{2(enol)}$) becomes slightly smaller than for peak B or the absorption edge ($\sim$ 291 eV). Considering that other hydrated forms of methylglyoxal contribute to the spectra in the energy range of peak B, this can indicate smaller abundance of the enol form compared to other hydrated forms in salt solutions. However, the addition of salts changes the shape of the background due to close-lying ionization continua of the Cl and S L-shells [11], and thus these changes remain inconclusive."**

*Line 274: This sentence states that C=O and C-H functional groups are found in the dihydrate form of methylglyoxal. For C=O this statement is false.*

**Authors' reply:**

**We agree that this should be corrected and we modify the sentence "They are found in the monohydrated and dihydrated forms of the compound and also from products of keto–enol tautomerism followed by aldol condensation reactions" (lines 274–275) to:**

**"They are found in the unhydrated and monohydrated forms of the compound and also in products of keto–enol tautomerism followed by aldol condensation reactions."**

*Technical Corrections:*

*Table 3: Why are excitation energies not listed for dehydrated methylglyoxal? The text states that they were calculated in line 136.*

**Authors' reply:**

**We agree that the calculated energies of dihydrated methylglyoxal should be discussed. We add in the caption of Table 3 the following sentence:**

**"For dihydrated methylglyoxal, our calculations did not identify any C 1s-$\pi$* excitations in the energy range of the recorded spectra and only the calculated IP are presented below."**

*Line 159: Are these differences of 0.1 – 0.2 eV significant, given that they are "within the photon energy resolution"? Or are they just random variation? If random, Table 4 should be moved to the SI section or eliminated from discussion.*

**Authors' reply:**

We agree that Table 4 could be placed in the SI. While the values are close to experimental accuracy, they show a systematic shift of 0.1 eV when salts are added. We move Table 4 to the SI along with the text "The absorption edges . . . concentration" (lines 158–161). We replace the above sentences with the following text:

**"The location of the absorption edge of aqueous solutions of pure glyoxal was at 289.6±0.1 eV (0.5 M) and 289.5±0.1 eV (1 and 2 M) and upon addition of inorganic salts, the absorption edge energy systematically increased by 0.1 eV in all cases. All values are however close to the experimental accuracy. All absorption edge energies can be found in the Supporting Information (Table S1)."**

*In Table 5, the functional group entry for peak B should also be labelled "enol", if I understand Figure 5 correctly. Also, no oscillator appears to line up with the C3 peak, so the origin of its assignment in Table 5 is unclear.*

**Authors' reply**:

Thank you for your suggestion. To be consistent with the notations in Figure 5, we change the labels of the peaks A and B in Table 5 to **"CH$_{2(enol)}$"** and **"C-OH(CH$_2$)$_{(enol)}$"**, respectively.

*Figures 6-8 have two deep pink lines in the legend that look identical, but not in the figure itself.*

**Authors' reply**:

Thank you for bringing this to our attention. We replace the purple dotted line of the legends in Figures 6, 7 and 8 that concern solutions with Na$_2$SO$_4$ with a purple dashed line which is more readily distinguished.

*Line 276: This sentence calls 1 – 2 M "low solute concentrations," which is somewhat misleading. It seems like the issue is rather the large signals of water.*

**Authors' reply**:

We agree that this should be modified. We rephrase the text "The oxygen K-edge spectra were similar to water, due to low solute concentrations and we only observed small changes in the case of mixtures with Na$_2$SO$_4$" (lines **276–277**) to:

**"The oxygen K-edge spectra for aqueous solutions were similar to**

those of pure water. Organic and inorganic solutes did not remarkably modify the water network at the studied concentrations, except in the case of $Na_2SO_4$ ....”

**Anonymous reviewer #4**

*In the introduction, there is no mention of relevant literature with regards to characterizing molecular structure and electrolyte-nonelectrolyte interactions in relevant systems. This is actually partially covered later in the discussion, but the work should be placed in context of prior art up front.*

**Authors' reply**:

**Thank you very much for your suggestion. We add in line 46 of the Introduction:**

**Recently, XAS has been used to study both solute–solute and solute–solvent interactions, including e.g. investigation of structure of methanol–water mixtures based on C and O K-edge XAS [6], quantification of sulfuric acid–water interaction using O K-shell and S L-edge XAS [8], and studies of ion-water interactions [10, 7].**

*These ionic strengths are quite low by atmospheric standards so there should be a caveat regarding the limitations of this study.*

**Authors' reply**:

**Thank you very much, we agree that this should be discussed. We add the following text in section "Atmospheric implications" and line 258:**

**The studied concentrations of glyoxal, methylglyoxal and inorganic salts are higher than their typical concentrations in cloud water, which have been estimated to be about five or more orders of magnitude lower [1, 5, 4, 9, 3]. Droplet evaporation, however, can lead to highly concentrated and supersaturated solutions, altering the chemical and optical properties of aerosol particles [3].**

*The authors introduce the "NEXAFS" later in the manuscript as if it were a different technique, but it's not clear that this is actually different from the technique used by the authors referred under broader term, "XAS".*

**Authors' reply**:

**Thank you very much for pointing that out. To avoid any confusion, we modify the sentence "applying near-edge X-ray absorption fine structure (NEXAFS)" in line 198, to:**

**"applying XAS in near-edge region"**

and we replace the term **NEXAFS** in line **200** with the more general term **"XAS"**.

*Apart from studying water (as cited by the authors), the O K-edge has generally not been found to be useful for characterizing specific structures of organic compounds (apart from indicating total oxygen content) in past work, and also seems to be the case here. The O K-edge does not really make a contribution to the main findings of the manuscript and should probably be summarized in a few sentence and removed otherwise (or placed in supplemental).*

**Authors' reply**:

**Thank you very much for the comment, we agree that the importance of the O 1s edge measurements in general was not clear. We believe that the results for O 1s are important from the perspective of both the solute and solvent. This aspect is now emphasized also in the introduction where we include recent studies utilising also O 1s XAS (cf. response to the first comment of reviewer #4). To clarify the significance of O 1s XAS results in our work, we replace the text "We do not observe any relative changes in spectral features as a function of concentration of the organic compound" in lines 237–238, by:**

**"This is most likely due to oxygen from solute molecules contributing mainly to the absorption above 536 eV, but for 2 M solutions of glyoxal and glycerol, there is also a small increase in the intensity after the pre-peak".**

**We also replace the first paragraph on page 14 "In conclusion, O K-edge spectra were found to be sensitive neither to the organic component at studied concentrations nor to addition of NaCl. However, addition of $Na_2SO_4$ affected the spectra as they started to resemble more pure $Na_2SO_4$ solution without any organics. Thus, no $Na_2SO_4$–organic interaction can be confirmed using this method" (lines 250–252) by:**

**"In conclusion, addition of organics and NaCl do not modify the overall structure of the measured O 1s XAS spectra. However, the presence of strongly hydrated $SO_4^{2-}$ anions leads to an observable effect on both the pre-peak and main peak regions. The effect on the shape was the same regardless of the identity of the organic compound in the solution, and we were not able to confirm any $Na_2SO_4$–organic interaction in the present study."**

and we add in line **278** the sentence:

**"The change in the shape of the spectra does not depend on the organic component. Thus, based on our study, the salting effects in water solutions of glyoxal and methylglyoxal upon addition of $Na_2SO_4$ would originate from changes in the structure of water by $SO_4^{2-}$ anions, rather than interactions with the organic."**

*Table 4 also seems superfluous and can be moved to the supplemental since it can be summed up concisely in the text.* **Authors' reply**:

Thank you for the suggestion. We move the table 4 to the Supplementary Information along with the text "The absorption edges ... concentration" (lines 158–161). We replace the above sentences with the following text in the main manuscript:

**"The location of the absorption edge of aqueous solutions of pure glyoxal was at 289.6±0.1 eV (0.5 M) and 289.5±0.1 eV (1 and 2 M) and upon addition of inorganic salts, the absorption edge energy systematically increased by 0.1 eV in all cases. All values are however close to the experimental accuracy. All absorption edge energies can be found in the Supporting Information (Table S1)."**

*Can the authors elaborate on how the measured absolute intensity or optical density calibration is used in the experiments?*

**Authors' reply**:

Thank you for the comment, we agree that text related to spectral intensity and calibration was not sufficiently clear. We therefore add the symbol $I_0$ in line **74**:

"During each measurement the incident radiation $I_0$ was monitored with a gold mesh placed before the liquid cell, so that the flux variations ($<1\%$) due to the top-up mode were removed".

We also modify the text "The thickness of the liquid layer ($x$) was not precisely estimated. In order to avoid additional uncertainty on our results, the intensities of the XA spectra are given in arbitrary units." (lines **78–79**) to:

**"The thickness of the liquid layer ($x$) was not precisely determined, and thus we give the intensities of the XA spectra in arbitrary units (arb. units)."**

In addition, we change the word "calibration" in line **93** to "energy calibration" and we rephrase the "ionization edge" in line **97** to "absorption edge".

*The way the units is presented is not entirely clear since the authors talk about measured beam intensity and absorption intensity (abs. = absorbance units?). Can authors clarify in one of the Figure captions?*

**Authors' reply**:

Thank you for bringing that to our attention. As explained in our response to the previous comment, the intensities of XAS are given in arbitrary units. We correct the typo in all the figures and relabel the y-axis from "abs.units" to "arb. units".

*The inline equation at the top of pg. 5 should be the Lambert formula since Beer's contribution comes from introducing the molar or number concentration to the linear attenuation coefficient of the material.*

**Authors' reply**:

Thank you. We change "Beer-Lambert formula" in line **72**, to "Lambert formula".

*The authors seem to use the term "salting in/out" to generally refer to non-ideal electrolyte-nonelectrolyte interactions instead of the consequence of their interactions, unless they are expecting a specific type of change in the spectra that was not made explicit in the manuscript.*

**Authors' reply**:

We agree that we should define the salting in and out phenomena and describe how these could affect our spectra. We therefore add in line **23** of the Introduction section the sentence:

"Salting in and out effects refer here to the increase or decrease in the solubility of the organic solute (glyoxal and methylglyoxal) in water due to the presence of a co-solute, in this case an inorganic salt ($NaCl$, $Na_2SO_4$), in the solution."

And we modify the text "in both carbon and oxygen K-edge spectra, we did not observe any effects ... demonstrated by Kurtén et al. (2014)", in lines **279–281**, to:

"In the carbon K-edge spectra we did not observe significant changes with addition of salts. Our observation excludes any significant organic-

**inorganic interactions that would change the abundances of different hydrated forms and does not reveal appearance of new species from such interactions. However, XAS might not be sufficiently sensitive to see additional changes to the hydrate formation equilibrium, as e.g. demonstrated computationally by Kurtén et al. (2014)."**

*Fig. 3: One of the vertical lines appear red while the other appears purple. It should be specified that it's these vertical lines that are scaled according to oscillator strengths and not the spectra (which are also drawn as lines).*

**Authors' reply**:

**We change the sentence "The lines have been scaled according to the calculated oscillator strengths" in captions of Fig. 3 and Fig. 5 to:**

**"The vertical lines have been scaled according to the calculated oscillator strengths."**

*Fig. 5. Since the overall fit is shown, can the authors display all the curves that are fitted (not just the Gaussian peaks)?*

**Authors' reply**:

**Thank you for the suggestion. We add in Fig. 5 the three curves that we fit in our spectrum at energies above 290 eV.**

*The authors talk about "IP of the C=O sites" (and for other groups) but would maybe more naturally be referred to as the "IP of the C=O moiety" in this context?*

**Authors' reply**:

**We change the word "sites" to "moieties" in lines 134, 135, 144, 145, 146, 147, 204 and in Table 3.**

*"SCF" is not defined.*

**Authors' reply**:

**Thank you, we agree that SCF should be defined. We add the definition in line 109:**

**"The core ionization and excitation energies were evaluated within the $\Delta$SCF (Self-Consistent Field) method . . . ".**

**References**

[1] Manabu Igawa, J. William Munger, and Michael R. Hoffmann. Analysis of aldehydes in cloud- and fogwater samples by hplc with a postcolumn reaction detector. *Environmental Science & Technology*, 23(5):556–561, 1989.

[2] Christopher J Kampf, Eleanor M Waxman, Jay G Slowik, Josef Dommen, Lisa Pfaffenberger, Arnaud P Praplan, André SH Prévôt, Urs Baltensperger, Thorsten Hoffmann, and Rainer Volkamer. Effective henry's law partitioning and the salting constant of glyoxal in aerosols containing sulfate. *Environmental science & technology*, 47(9):4236–4244, 2013.

[3] Alex K. Y. Lee, Ran Zhao, Richard Li, John Liggio, Shao-Meng Li, and Jonathan. P. D. Abbatt. Formation of light absorbing organo-nitrogen species from evaporation of droplets containing glyoxal and ammonium sulfate. *Environmental Science & Technology*, 47(22):12819–12826, 2013. PMID: 24156773.

[4] Kiyoshi Matsumoto, Shunji Kawai, and Manabu Igawa. Dominant factors controlling concentrations of aldehydes in rain, fog, dew water, and in the gas phase. *Atmospheric Environment*, 39(38):7321–7329, 2005.

[5] J William Munger, Daniel James Jacob, Bruce C Daube, LW Horowitz, WC Keene, and Brian G Heikes. Formaldehyde, glyoxal, and methylglyoxal in air and cloudwater at a rural mountain site in central virginia. *Journal of Geophysical Research: Atmospheres*, 100(D5):9325–9333, 1995.

[6] Masanari Nagasaka, Kenji Mochizuki, Valentin Leloup, and Nobuhiro Kosugi. Local structures of methanol–water binary solutions studied by soft x-ray absorption spectroscopy. *The Journal of Physical Chemistry B*, 118(16):4388–4396, 2014.

[7] Masanari Nagasaka, Hayato Yuzawa, and Nobuhiro Kosugi. Interaction between water and alkali metal ions and its temperature dependence revealed by oxygen k-edge x-ray absorption spectroscopy. *J. Phys. Chem. B*, 121(48):10957–10964, 2017.

[8] Johannes Niskanen, Christoph J Sahle, Iina Juurinen, Jaakko Koskelo, Susi Lehtola, Roberto Verbeni, Harald Müller, Mikko Hakala, and Simo Huotari. Protonation Dynamics and Hydrogen Bonding in Aqueous Sulfuric Acid. *J. Phys. Chem. B*, 119(35):11732–11739, August 2015.

[9] Dominik van Pinxteren, Khanneh Wadinga Fomba, Stephan Mertes, Konrad Müller, Gerald Spindler, Johannes Schneider, Taehyoung Lee, Jeffrey L Collett, and Hartmut Herrmann. Cloud water composition during hcct-2010: Scavenging efficiencies, solute concentrations, and droplet size dependence of inorganic ions and dissolved organic carbon. 2016.

[10] Iradwikanari Waluyo, Dennis Nordlund, Uwe Bergmann, Daniel Schlesinger, Lars GM Pettersson, and Anders Nilsson. A different view of structure-making and structure-breaking in alkali halide aqueous solutions through x-ray absorption spectroscopy. *The Journal of Chemical Physics*, 140(24):244506, 2014.

[11] JJ Yeh and I Lindau. Atomic subshell photoionization cross sections and asymmetry parameters: $1 \leq z \leq 103$. *Atomic data and nuclear data tables*, 32(1):1–155, 1985.

[12] Ge Yu, Amanda R Bayer, Melissa M Galloway, Kyle J Korshavn, Charles G Fry, and Frank N Keutsch. Glyoxal in aqueous ammonium sulfate solutions: products, kinetics and hydration effects. *Environmental science & technology*, 45(15):6336–6342, 2011.

---

## Author Response (AR2)

January 6, 2021

Dear Editor,

Thank you for your comments on our manuscript. Please find below our responses to your comments.

**p. 1 line 10 I think it should be "indicating that XAS in the" rather than "indicating that the XAS in the"**

Thank you, we have corrected the text in line 10.

p. 2 line 35, for ammonia please also cite Galloway et al. 2009

We have added the reference in line 35.

p. 2 line 41 I think this should be "has a lower value" and not "has lower value"

We have corrected the text in line 41.

**p. 3. Line 61-62 Can a statement be made on the purity of the glyoxal and methylglyoxal solutions?**

It is stated that the aqueous solution of glyoxal has acidity due to CHOCOOH while we cannot provide a statement about the solution of methylglyoxal. We modify the text in lines 61–62: "Glyoxal and methylglyoxal were delivered as 40 wt% aqueous solutions. Stated acidity of the aqueous glyoxal (glyoxylic acid, CHOCOOH) is maximum 2%. Glycerol was a viscous liquid of 99.5% purity."

p. 3 line 76 Perhaps better "pumped through the liquid cell with a flow rate  $\dots$ "

Thank you for the suggestion. We have changed the text in lines 75–76 to: "The liquid sample can be pumped through the liquid cell with adjustable flow rate (Cole–Parmer Masterflex L/S pump system)."

**p. 4 Table 1. Could the authors please double check the solubility of NaCl. It seems a little high, as I thought it was around 36g of NaCl per 100g of water with little temperature dependence.**

We agree that the given solubility of NaCl is high and we change the value to 36 g/100 g [1].

**p. 5 line 96. Perhaps better "The C K-edge spectra of pure water were subtracted from the C K-edge spectra of the samples to ..."**

Thank you, we rephrase the sentence in line 96 as you have suggested.

p. 9 figure 3. What are the energies for the oligomeric C-O-C acetal. This could have a high enough concentration for glyoxal to be observed see figure S9 in Yu et al. 2011 especially as those were only 1M solutions?

p. 10 line 182-183. Regarding the absorption edge energy increase with decreasing concentration of glyoxal (2M to 1M) and with addition of salt. Could this be explained by a decrease of the oligomer to monomer ratio with decreasing glyoxal concentration and increasing salt concentration as hypothesized by Yu et al. 2011? I think this is consistent with the results presented here?

Unfortunately, the absorption energies of oligomers have not been calculated. However, we agree that this should be discussed further and that the changes on the absorption edge energies could be related to the abundance of oligomers. We have changed the text in line 181 to: "Kua et al. (2008) [2] and Yu et al. (2011) [3] identified potential oligomers containing acetal groups in aqueous glyoxal. Here, we did not calculate absorption energies of oligomers but based on the C 1s absorption energies of similar O-C-OH functional groups found in saccharides (Gainar et al., 2015 [4]), we estimate that in oligomers acetal moieties would have higher absorption energies compared to C-OH moieties. Their contribution would therefore be embedded in the broad continuum part of the XAS spectrum. The observed increase in absorption edge energy with decreasing concentration of glyoxal and addition of salts could be related to changes in the degree of oligomerisation. This is in line with the study of Yu et al. (2011) [3] who observed deoligomerization with addition of Na2SO4 and NaCl."

p. 15 line 270 -271 p. 16 line 299-300. Yu et al. did not conclude that glyoxal exists entirely in its fully hydrated form, as they were able to observe aldehyde 1H NMR signals. The NMR intensities of the aldehydic hydrogen was very low but the authors were able to show that upon addition of salts the dihydrate to monohydrate signal increased, see Table 1 in their paper.

Thank you very much for pointing this out. We have changed the text in lines 270–271 to: "Here, we confirm experimentally that glyoxal, to the sensitivity of our measurements, exists strongly in its fully hydrated form in aqueous solution, in agreement with previous studies [3, 5, 2]" and we remove the sentence in lines 299–300: "However, our finding that glyoxal exists entirely in its covalently bond dihydrate form in aqueous solution is in line with the results of Yu et al. (2011)."

**p. 16 line 288, please add "to the sensitivity of our measurements"**

Thank you, we have added the text in line 288.

Supplement p 1. Line 28. Should this be aldehydic rather than ketonic for glyoxal, as it has only aldehyde groups and no keto groups? What is the purity of the solution. Glyoxal has a very high Henry's law constant, could contaminations of higher vapor pressure glyoxylic or glycolic acid cause the feature at 288.7eV?

Thank you for bringing the typo in our attention. We change the word "ketonic" to "aldehydic" in line 28 of the Supporting Information. It is estimated that the solution acidity due to glyoxylic acid is maximum 2%. We add in line 33 of the SI: "indicating possible presence of acid (as CHOCOOH,  $\sim 2\%$  according to manufacturer Wako) impurities in the solution."

Sincerely, on the behalf of the authors,

**Georgia Michailoudi**

\*References

- D. R. Lide. CRC Handbook of Chemistry and Physics, 85th Edition. CRC Handbook of Chemistry and Physics. Taylor & Francis, 2004.
- [2] J. Kua, S. W. Hanley, and D. O. De Haan. Thermodynamics and Kinetics of Glyoxal Dimer Formation: A Computational Study. J. Phys. Chem. A, 112(1):66–72, January 2008.
- [3] G. Yu, A. R. Bayer, M. M. Galloway, K. J. Korshavn, C. G. Fry, and F. N. Keutsch. Glyoxal in aqueous ammonium sulfate solutions: Products, kinetics and hydration effects. *Environ. Sci. Technol.*, 45(15):6336–6342, 2011.
- [4] A. Gainar, J. S. Stevens, C. Jaye, D. A. Fischer, and S. L. M. Schroeder. Nexafs sensitivity to bond lengths in complex molecular materials: a study of crystalline saccharides. J. Phys. Chem. B, 119(45):14373-14381, 2015.
- [5] Malik M. and Jeffrey J. A. Temperature dependent near-uv molar absorptivities of glyoxal and gluteraldehyde in aqueous solution. *Spectrochim. Acta*, *Part A*, 56(14):2653 – 2658, 2000.